# Cloning, Secretory Expression and Characterization of a Unique pH-Stable and Cold-Adapted Alginate Lyase

**DOI:** 10.3390/md18040189

**Published:** 2020-04-01

**Authors:** Zhi-Peng Wang, Min Cao, Bing Li, Xiao-Feng Ji, Xin-Yue Zhang, Yue-Qi Zhang, Hai-Ying Wang

**Affiliations:** 1Key Laboratory of Sustainable Development of Polar Fishery, Ministry of Agriculture and Rural Affairs, Yellow Sea Fisheries Research Institute, Chinese Academy of Fishery Sciences, Qingdao 266071, China; spirit87@163.com (Z.-P.W.);; 2Marine Science and Engineering College, Qingdao Agricultural University, Qingdao 266109, China; caominjiayou@163.com (M.C.); qdaulb@163.com (B.L.); zxy17852840297@163.com (X.-Y.Z.); zyq15169506569@163.com (Y.-Q.Z.); 3Laboratory of Enzyme Engineering, Yellow Sea Fisheries Research Institute, Qingdao 266071, China; 4Laboratory for Marine Drugs and Bioproducts, Qingdao National Laboratory for Marine Science and Technology, Qingdao 266071, China

**Keywords:** alginate lyase, cold-adapted, pH-stable, NaCl-independent

## Abstract

Cold-adapted alginate lyases have unique advantages for alginate oligosaccharide (AOS) preparation and brown seaweed processing. Robust and cold-adapted alginate lyases are urgently needed for industrial applications. In this study, a cold-adapted alginate lyase-producing strain *Vibrio* sp. W2 was screened. Then, the gene *ALYW201* was cloned from *Vibrio sp.* W2 and expressed in a food-grade host, *Yarrowia lipolytica*. The secreted Alyw201 showed the activity of 64.2 U/mL, with a molecular weight of approximate 38.0 kDa, and a specific activity of 876.4 U/mg. Alyw201 performed the highest activity at 30 °C, and more than 80% activity at 25–40 °C. Furthermore, more than 70% of the activity was obtained in a broad pH range of 5.0–10.0. Alyw201 was also NaCl-independent and salt-tolerant. The degraded product was that of the oligosaccharides of DP (Degree of polymerization) 2–6. Due to its robustness and its unique pH-stable property, Alyw201 can be an efficient tool for industrial production.

## 1. Introduction

Brown seaweed (Phaeophyceae) is considered one of the most promising bioresources due to its high growth rate and huge biomass [1]. Multiple bioactive substances, such as alginate, laminarin, fucoidan, fucoxanthin, mannitol, polyphenol, and phytohormones, are abundant in brown seaweed and are applied in different industries [2,3]. Among these bioactive substances, alginate accounts for approximately 22%–44% (w/w) of seaweed biomass and forms the major component of the cell wall [4,5]. Bioactive substance extractions require the degradation of alginate to destroy the structure of the cell wall.

Alginate is a kind of linear polymer, composed of β-d-mannuronate (M) and α-l-guluronate (G) as the monomer units. The monomers are combined by 1,4-glycosidic linkages in different orders, forming polyguluronate blocks (PolyG), polymannuronate blocks (PolyM), and heteropolymer blocks (PolyMG). Alginate lyases, belonging to the family of polysaccharide lyases (PLs), cut the 1,4-glycosidic linkages to form oligosaccharides or monosaccharides. Alginate lyase is adopted to hydrolyze alginate into alginate oligosaccharides (AOS) under relatively mild and controllable conditions. AOS is reported to have antioxidant, anti-tumor, anti-bacterial, and anti-hyperglycemic properties, and to have potential applications in health-related fields [3,6,7]. Furthermore, AOS has been proven to show a prominent promoting effect on plant root development [8,9].

Various alginate lyases have purified and characterized form different bacteria, such as *Alteromonas*, *Pseudoalteromona*, *Thalassomonas*, *Flavobacterium*, *Bacillus*, *Nitratiruptor*, *Agarivorans*, and *Vibrio* [10,11,12,13,14,15]. The biochemical properties of these enzymes have been determined; the coding genes have been cloned, modified, and expressed [10,11,12,13,14,15]. Among them, cold-adapted alginate lyases have aroused wide attention [15]. Although industrial enzymes generally require thermostability, several advantages can be achieved when the hydrolysis was performed during the cultivation of a recombinant strain with cold-adapted alginate lyases. Under lower temperatures, the instability of some bioactive substances, such as phytohormones, is weakened; thus, more active substances are retained during the extraction from brown seaweed, accompanied by the biocatalytic alginate-degrading process [7]. However, few reported cold-adapted alginate lyases meet industrial requirements, due to the limited application conditions, low activity, and weak stability [15,16,17,18,19]. However, robust and cold-adapted alginate lyases are urgently needed for industrial applications.

In this study, a novel alginate lyase-producing strain, *Vibrio sp.* W2, was screened at a low temperature. Then, the alginate lyase-encoding gene *ALYW201* was cloned and expressed in a food-grade and eukaryotic host, *Yarrowia lipolytica* (Fungi). The secreted alginate lyase Alyw201 showed obvious robustness, excellent pH stability, and NaCl-independent and salt-tolerant properties. These special features suggest that Alyw201 can be an efficient tool for industrial production.

## 2. Results

### 2.1. Vibrio sp. W2 Has the Ability of Alginate Degradation at a Low Temperature

Abalone feed on different kinds of natural seaweeds or seaweed-containing fodder. Thus, alginate-degrading microbial communities should be present in the viscera of abalone [20]. After isolation and purification, one hundred and six alginate-degrading strains could grow on the ASC (Alginate as sole carbon source) plates at 25 °C. After separate staining by Lugol’s iodine solution and CaCl_2_ solution, the degraded alginate part on the plate showed a different color to the part with the original alginate. By comparing the sizes of the transparent circles around the strains, the relative level of alginate lyase activity at 25 °C could be evaluated. As shown in Figure 1, strain W2 generated the biggest transparent circle on the plate, indicating that this strain secreted extracellular and cold-adapted alginate lyase with the highest activity. 

On the ASC plate, the colony of strain W2 was smooth, milky yellow, opaque, and round with a slight convex; the edges of the colony were neat (Figure 2a). Through microscope observation, the strain had the characteristics of bacteria. To identify the strain, the 16S rDNA of strain W2 was sequenced. After sequence comparison by BLAST (Basic Local Alignment Search Tool), it was found that the 16S rDNA sequence was most similar to identified strain *Vibrio algivorus* (98.05%). However, by clustering analysis, strain W2 was on the same branch as *Vibrio casei* (Figure 2b). Exactly to which specific species the strain W2 belongs to requires further verification. Comprehensively considering the BLAST and phylogenetic tree results, W2 belongs to the genus *Vibrio* and was designated as *Vibrio* sp. W2. 

### 2.2. Bioinformatics Analysis of the Alginate Lyase Alyw201

To clone the gene of cold-adapted alginate lyase in strain *Vibrio sp.* W2, the genome DNA was sequenced. The sequence analysis showed that a putative gene encoding alginate lyase existed. The open reading frame (ORF) consists of 1047 bp and encodes a protein of 384 amino acids. The gene was designated as *ALYW201*, and further bioinformatics analysis was performed. The theoretical isoelectric point (pI) was 5.53, and the molecular weight (Mw) was 36.4 kDa. The first 20 amino acids of Alyw201 were predicted to be the signal peptide, consistent with the secretion characteristic. By BLAST on the National Center for Biotechnology Information (NCBI), Alyw201 was shown to have one conserved domain, belonging to the polysaccharide lyase (PL) family 7 and alginate lyase superfamily 2.

To further confirm the attribution of Alyw201, phylogenetic trees were constructed according to the amino acid sequences of Alyw201 and other reported alginate lyases. As shown in Figure 3, Alyw201 was clearly grouped in the PL7 family, and was gathered on the same branch as two PL7 family alginate lyases, one from *Vibrio halioticoli* (AAF22512.1) and the other from *Zobellia galactanivorans* (CAZ95239.1) [14]. Furthermore, multiple sequence alignment was carried out between Alyw201 and five well-characterized alginate lyases of the PL7 family. The five alginate lyases included alginate lyase from *Klebsiella pneumoniae* (Accession: AAA25049.1), alginate lyase from *Agarivorans* sp. L11 (Accession: AJO61885.1), alginate lyase from *Saccharophagus degradans* 2–40 (Accession: ABD81807.1), alginate lyase from *Marinimicrobium* sp. (Accession: QGU34249.1), and alginate lyase from *Vibrio* sp. (Accession: ASA33935.1) [21,22,23,24]. The results show that Alyw201 contains the typical conserved motifs of the PL7 family, such as “RT/SELRE,” “QIH,” and “MYFKAG” (Figure 4). These amino acids have been verified by site-specific mutagenesis to bind the alginate substrate or to determine the catalytic activity [25]. Commonly, isoleucine (I) in the “QIH” residue recognizes the substrate; alginate lyases containing the “QIH” motif showed a preference toward polyG blocks [21,22,23,24,25]. It was reported that the five alginate lyases in Figure 4 indeed showed a polyG preference. On the contrary, other alginate lyases containing the “QVH” motif preferred to degrade polyM blocks [26,27]. Thus, Alyw201 may be also a polyG-preferred alginate lyase.

### 2.3. Secretory Expression and Purification of Alyw201

In previous studies, most alginate lyases were expressed in *Escherichia coli*. Nevertheless, using *E. coli* for the industrial production of alginate lyases is limited, due to its weak secretory ability, endotoxin synthesis, and presence of cell wall pyrogens [28]. In the present study, Alyw201 was expressed in the yeast *Y. lipolytica*, a widely utilized heterologous host that affords remarkable extracellular secretion [29,30]. As shown in Figure 5a, after incubation in GPPB (Glucose in phosphate buffer) medium for 60 h, the alginate lyase activity in the recombinant strain Y23 achieved 64.2 U/mL, with a biomass of 16.2 g/L. This activity was almost twice of that of the wild strain (38.3 U/mL). The Alyw201 protein was purified from the supernatant and was analyzed by Sodium dodecyl sulfate polyacrylamide gel electrophoresis (SDS-PAGE). As shown in Figure 5b, a distinct single band appeared in the lane; the Mw of Alyw201 was approximately 38.0 kDa. However, the theoretical Mw of secreted protein with the His-tag should be 37.2 kDa. The discrepancy with the Mw determined by SDS-PAGE may be attributed to the potential glycosylation at the amino acid region (NRTD). The glycosylation may affect the structure and inhibit docking with substrate. The specific activity of purified Alyw201 toward sodium alginate was 876.4 U/mg, while those toward polyG blocks and polyM blocks were 1281.2 U/mg and 713.6 U/mg, respectively. Thus, the prediction of polyG preference by bioinformatics analysis was verified.

### 2.4. Temperature and pH Properties of Alyw201

The enzymatic properties of the purified Alyw201 were investigated. As shown in Figure 6a, the catalyzing activity of Alyw201 was highest at 30 °C; more than 80% of the highest activity was detected at 25–40 °C. As the temperature rose above 45 °C, the activity declined dramatically. Meanwhile, the catalyzing activity was 72.9% and 38.4% of the highest activity, at a low temperature, 10 °C and 20 °C, respectively. At a temperature below 40 °C, Alyw201 was remarkably stable; approximately 80% of the activity was maintained after incubation at 40 °C for 2 h. However, most of the activity was lost above 45 °C. Most of the reported alginate lyases showed their highest activity at about 40 °C [21,22,23,24,25]; meanwhile, the cold-adapted alginate lyases performed their highest catalytic activities at less than 35 °C, and commonly performed more than 50% of the highest activity at 20 °C (Table 1). Compared with the cold-adapted alginate lyases in Table 1 and Figure 6b, Alyw201 showed higher activity at 20 °C, and possessed better thermostability. Thus, the results indicate that Alyw201 is a typical and robust cold-adapted alginate lyase. Reduced consumption energy, risk of contamination, and inactivation difficulty can be achieved during the catalytic process of cold-adapted alginate lyases. Alyw201 may be a new catalytic tool for potential industrial applications [16].

As shown in Figure 7a, the activity of Alyw201 was highest at pH 8.0, and was above 80% from pH 6.0 to 10.0. Furthermore, more than 70% of the activity was obtained in a pH range of 5.0–10.0 after a 12-h incubation. Especially, in the whole pH range (3.0–11.0) detected in this study, more than 40% of the activity remained after the incubation (Figure 7b). Generally, alginate lyases from bacteria tend to catalyze the hydrolysis reaction in a neutral and narrow pH range [15,16,17,24,26]. Compared with the cold-adapted alginate lyases in Table 1, Alyw201 performed the catalyzing activity in a much broader pH range; the pH stable range of Alyw201 was also much broader. For example, the pH-stable ranges of TsAly6A and AlgNJU-03 were just 6.6–8.95 and 6.0–9.0, respectively [16,24]. Indeed, the pH range of Alyw201 with high activity and good stability was even broader than those of AlgNJ04, a well-recognized, pH-stable, and mesophilic alginate lyase [25]. This new alginate lyase, Alyw201, with a unique pH-stable property could facilitate the extraction and biotransformation of various bioactive substances in brown seaweed. 

### 2.5. Effects of Ions on the Activity of Alyw201

As shown in Figure 8a, at a concentration of 1 mM, no obvious inhibiting effect was observed in the presence of the metal ions. The activity of Alyw201 containing Mn^2+^ was 238.1% as high as that of the control; the relative activity of Alyw201 containing Co^2+^ was 195.0%. However, with SDS and Ethylenediamine tetraacetic acid (EDTA) at a concentration of 1 mM, the relative activity was still reduced to 60.7% and 15.2%, respectively. Critical inhibition by EDTA has also been found in several reported alginate lyases [19]. When metal chelator EDTA was added to the reaction mixture, the activity of alginate lyase AlyGC decreased with the increase of EDTA concentration, and 0.2 mM EDTA completely abolished the enzyme activity [19]. A combined structural and mutational analysis indicates that Ca^2+^, which is far away from the active center, is involved in catalysis and plays an important role in stabilizing the structure. In this study, it was also suspected that EDTA chelates the metal ions crucial for the function and structure of Alyw201 [19]. At a concentration of 10 mM, the effects were quite different. Only Fe^3^^+^, Al^3^, Ba^2+^, Cu^2+^, SDS, and EDTA showed obvious inhibiting effects on the activity of Alyw201, while Co^2+^ still strongly activated the activity. The activity of Alyw201 containing Co^2+^ was 175.8% as high as that of the control. Thus, Alyw201 showed good metal ion tolerance, and Co^2+^ can be a good activating agent for Alyw201 [15,16,17,18,19].

In addition, as shown in Figure 8b, NaCl slightly activated Alyw201 at a concentration of 0–2.0 M, and slightly inhibited the activity at a concentration of 2.0–3.0 M. With 0.75 M NaCl, the activity of Alyw201 reached the highest (124.7%). For most alginate lyases, NaCl of a certain concentration is essential for the enzyme activation; the activity was generally quite a lot lower in the absence of NaCl. The activity of AlgM4 was increased by about seven times at 1 M NaCl [32]; Aly08 was also greatly enhanced by NaCl and the activity reached about eight times higher at 0.3 M NaCl than the activity in the absence of NaCl [17]. By contrast, Alyw201 was not so NaCl-dependent. Enhanced by NaCl, the highest activity of Alyw201 was just 124.7%. Meanwhile, the activity of Alyw201 was just slightly reduced with NaCl of high concentrations. Thus, Alyw201 showed stable activity, not obviously affected by the concentration of NaCl. Due to the NaCl-independent and salt-tolerant properties, Alyw201 was competent for a variety of specific processing requirements.

### 2.6. Product Analysis of Alyw201

Alginate solution was hydrolyzed by Alyw201 for 40 min. As shown in Figure 9a, the monitored viscosity decreased drastically in the first 10 min, and was stable at a low level after 20 min. Meanwhile, the absorbance at 235 nm continued to increase until 35 min. When the viscosity and absorbance were both steady, the final product was detected by negative-ion ESI–MS (Electrospray ionization mass spectrometry) (Figure 9b). A series of oligosaccharides were detected, such as disaccharides (351.05 *m/z*), trisaccharides (527.08 *m/z*), tetrasccharides (703.11 *m/z*), pentasaccharides (879.13 *m/z*), and hexaose (1055.15 *m/z*) Thus, Alyw201 is an endo-alginate lyase based on the distribution of the degradation products. The cold-adapted alginate lyases Algb and AlgNJU-03 produced oligosaccharides of DP (Degree of polymerization) 2–5, similar to the products degraded by Alyw201 [24,25,26,27,28,29,30,31]. This enzyme may be used for preparing oligosaccharides of lower molecular weights with potential pharmaceutical applications.

## 3. Materials and Methods

### 3.1. Materials, Strains, and Mediums

Sodium alginate was purchased from Bright Moon Seaweed Group (Qingdao, China). PolyM, and PolyG were purchased from Qingdao BZ Oligo Biotech Co., Ltd (Qingdao, China). The uracil mutant *Y. lipolytica* URA-strain and expression vector pINA1312 were kindly provided by Zhenming Chi, Ocean University of China. The alginate lyase-producing strains were cultivated in sole-carbon source (ASC) medium, containing 10 g/L sodium alginate, 5 g/L (NH4)_2_SO_4_, 1.0 g/L MgSO_4_·7H_2_O, 0.1 g/L FeSO_4_, and 20 g/L agar, prepared using seawater. *Y. lipolytica* URA-transformants were screened on a Yeast nitrogen base (YNB) plate, containing 1.7 g/L yeast nitrogen base without amino acids, 10.0 g/L glucose, 5.0 g/L (NH_4_)_2_SO_4_, and 25.0 g/L agar [29]. GPPB medium was used for recombinant enzyme production and contained 30.0 g/L glucose, 1.0 g/L (NH4)_2_SO_4_, 2.0 g/L yeast extract, 2.0 g/L KH_2_PO_4_, 3.0 g/L K_2_HPO_4_, and 0.1 g/L MgSO_4_·7H_2_O, with a pH of 6.8 [30].

### 3.2. Screening Alginate Lyase-Producing Strains at Low Temperature

Abalone viscera were collected and milled. The samples were spread on ACS plates. The plates were incubated at 25 °C for 48 h. Then, 106 single colonies were selected and transferred to two new ACS plates. The new ACS plates were separately stained by Lugo’s iodine solution and 0.5% (w/v) CaCl_2_ solution for 1 h. The colony of strain W2 was transferred into ASC liquid medium and inoculated for 48 h at 25 °C. The culture was centrifuged at 5000× *g* to determine the alginate lyase activity in the supernatant. The alginate lyase activity assay was performed at 35 °C using 0.5% (w/v) alginate solution as the substrate, as described in a previous study [33]. Enzyme activity was determined by increasing A235 as the hydrolysis reaction formed unsaturated double bonds. One unit (U) of enzyme activity was defined as the amount of enzyme required to increase A235 by 0.1 per minute, under the above conditions.

### 3.3. Strain Identification

16S rDNA of strain W2 was amplified through PCR reaction using general primers 27F (5’-AGAGTTTGATCCTGGCTCAG-3’) and 1492R (5’-TACCTTGTTACGACTT-3’). 16S rDNA was then sequenced and compared with other 16s rDNA sequences by BLAST. The neighbor-joining phylogenetic tree was generated based on the closely related sequence using MEGA version 7.0 (Center for Evolutionary Medicine and Informatics, The Biodesign Institute, Tempe, AZ, USA).

### 3.4. Bioinformatics Analysis of the Alginate Lyase Alyw201

To obtain the coding sequence of alginate lyase, the genomic DNA of strain W2 was sequenced and annotated (Novogene, Beijing, China). HMMER3 was used to compare the gene protein sequence with the CAZy (Carbohydrate-Active enZYmes) database to obtain the corresponding carbohydrate-active enzyme annotation information. The filter condition is E-value <1e-5. The sequence analysis showed that a putative gene encoding alginate lyase existed, with an ORF of 1047 bp (Accession: MT232847). The signal peptide was analyzed using the SignalP 4.1 server (http://www.cbs.dtu.dk/services/SignalP-4.1/). Domain analysis was performed in the Conserved Domain Database (https://www.ncbi.nlm.nih.gov/cdd). The theoretical pI and Mw were predicted online (http://web.expasy.org/compute_pi/). The neighbor-joining phylogenetic tree was generated based on the reported alginate lyases using MEGA version 7.0. Multiple sequence alignment was performed among the characterized PL7 family alginate lyases using DNAMAN 6.0 (Lynnon Corporation, QC, Canada).

### 3.5. Secretory Expression and Purification of Alyw201q

After codon optimization, the *ALYW201* gene with the XPR2 signal peptide gene was synthesized (Synbio Technologies, Suzhou, China). The optimized sequence was with an average GC content of 50% codon and an adaptable Index (CAI) of 0.84, without medium or large hairpins. The synthesized DNA fragment was transformed into the URA-strain [30]. After an 84 h cultivation in GPPB liquid medium at 30 °C, the alginate lyase activities of the positive transformants were detected. The recombinant strain Y23 showed the highest extracellular activity. During the fermentation period of Y23 at flask, the alginate lyase activity and the biomass were detected every 12 h. All of the data were collected in triplicate.

The supernatant of strain Y23 was adjusted to pH 7.5 and then applied to a Ni-NTA sepharose column (GE Healthcare, Chicago, IL, USA). The Alyw201 enzyme was attached to the gel and then washed off with imidazole solution. The purity and Mw of Alyw201 were verified by SDS-PAGE on 12% (w/v) gel. The catalyzing activities of Alyw201 were performed by using polyM, polyG, and alginate as substrates, respectively, to investigate the substrate preference.

### 3.6. Effects of Temperature and pH on Alyw201 Activity and Stability

The Alyw201-catalyzing hydrolysis reaction in 10 mM glycine–NaOH buffer of pH 8.0 was performed at temperatures ranging from 10 to 60 °C. The activities were detected to determine the best reaction temperature. To evaluate the thermal stability of Alyw201, purified Alyw201 was first incubated at temperatures ranging from 10 to 60 °C for 12 h. Then, the remaining activity was measured at 35 °C. The alginate solution as substrate was prepared in 10 mM buffers with different pH levels (Na_2_HPO_4_–citric acid, pH3.0–8.0; glycine–NaOH, pH 8.5–11.0). The enzymatic activity was assayed using these alginate solutions to determine the optimal reaction pH level. pH stability was detected by estimating the remaining enzyme activity after incubating for 12 h at 4 °C in buffers with different pH levels. All reactions were performed in triplicate.

### 3.7. Effects of Chemical Compounds, Metal Ions, and NaCl on Alyw201 Activity 

First, the mother liquors of the chemical compounds and metal ions were prepared. Then, the liquors were added to the alginate solutions with final concentrations of 1 mM and 10 mM, respectively. The Alyw201-catalyzing reactions were performed at 35 °C to measure the effects of these substances on activity. Meanwhile, Alyw201-catalyzing reactions were also performed in alginate solution at 35 °C with different concentrations of NaCl (0–3.0 M). The reaction in the original alginate solution containing no extra substance was used as a control. All reactions were performed in triplicate.

### 3.8. Analysis of Alyw201 Reaction Products

To obtain the completely degraded products from alginate, purified Alyw201 (40 U) was added into 10 mL of 0.5% (w/v) alginate solution at pH 8.0. The reaction was performed at 35 °C for 40 min. The viscosity and the absorbance at 235 nm were detected every 5 min. When the viscosity and the absorbance were both steady, the degraded products were desalted and investigated by ESI-MS (Thermo Fisher ScientificTM Q ExactiveTM Hybrid Quadrupole-OrbitrapTM, Waltham, MA, USA) to determine the degree of polymerization (DP). The oligosaccharides were detected in a positive-ion mode using the following settings: ion source voltage, 4.5 kV; capillary temperature, 275–300 °C; Tube lens, 250 V; sheath gas, 30 arbitrary units (AU); scanning the mass range, 150–2000 *m/z* [24].

## 4. Conclusions

In this study, a new typical cold-adapted alginate lyase was cloned, expressed extracellularly, and characterized. Alyw201 performed its highest activity at 30 °C, and performed more than 80% of its activity at 25–40 °C. Furthermore, more than 70% of activity was obtained in a broad pH range of 4.0–10.0. Alyw201 was also independent of NaCl concentration and salt tolerant. The degraded product was that of the oligosaccharides of DP2–DP6. Due to its robustness and its unique pH-stable property, Alyw201 can be an efficient tool for industrial applications.

## Figures and Tables

**Figure 1 marinedrugs-18-00189-f001:**
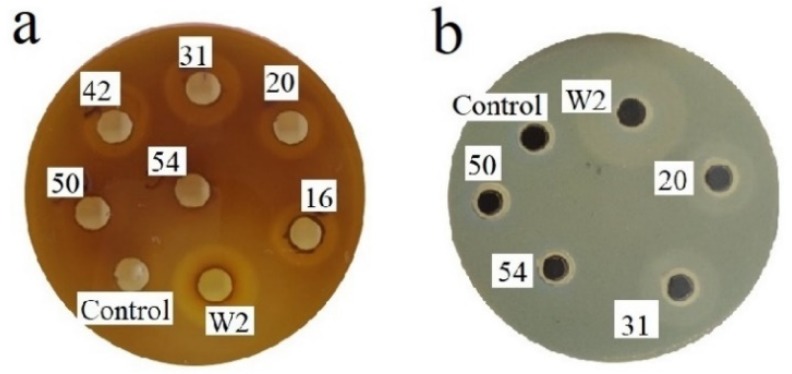
Alginate-degrading strains on the ASC plates after separate staining by Lugo’s iodine solution (**a**) and CaCl_2_ solution (**b**).

**Figure 2 marinedrugs-18-00189-f002:**
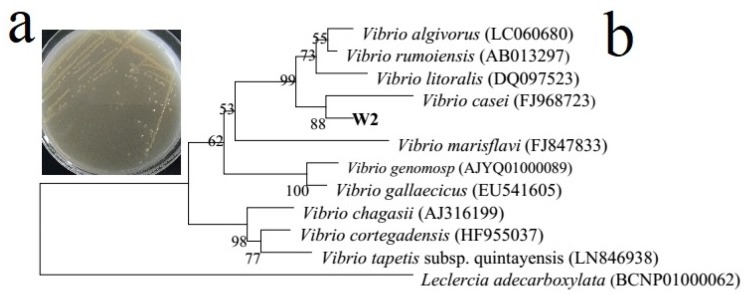
(**a**) The colony morphology of strain W2. (**b**) The neighbor-joining phylogenetic tree generated based on the 16S rDNA gene sequences. Branch-related numbers are bootstrap values (confidence limits).

**Figure 3 marinedrugs-18-00189-f003:**
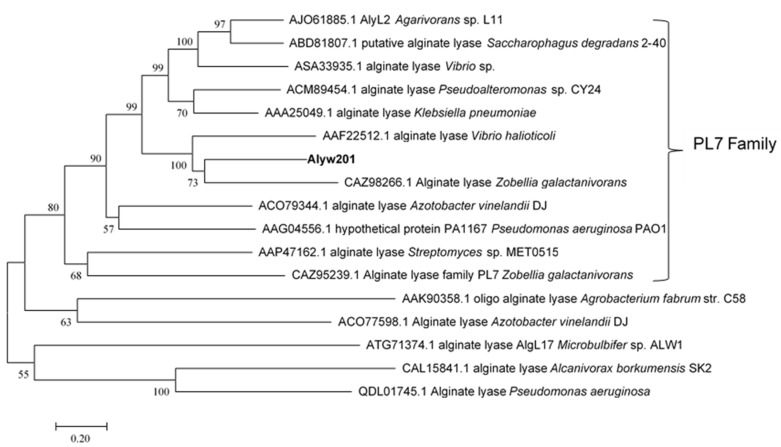
The neighbor-joining phylogenetic tree generated based on the amino acid sequences of the reported alginate lyases by the neighbor-joining method.

**Figure 4 marinedrugs-18-00189-f004:**
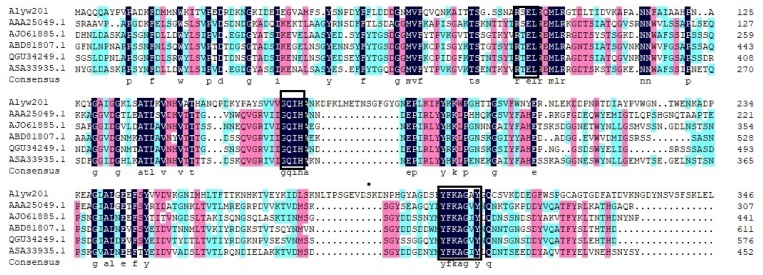
Multiple sequence alignments of Alyw201 and five well-characterized alginate lyases of the PL7 family. The conserved amino acid regions are marked in the black boxes.

**Figure 5 marinedrugs-18-00189-f005:**
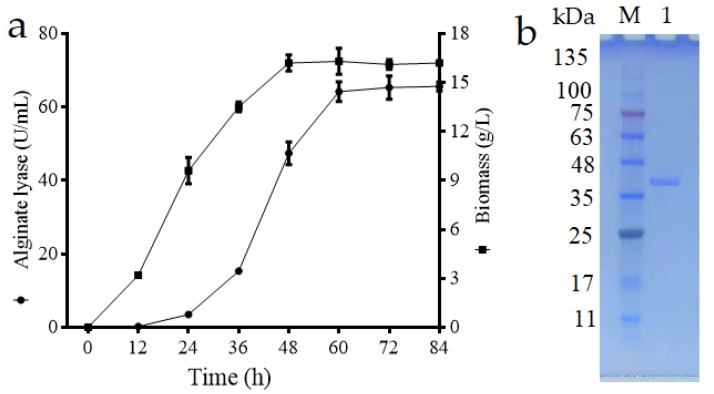
(**a**) Time curve of Alyw201 activity secreted into the medium. Data are given as means ± standard deviation, *n* = 3. (**b**) Analysis of Alyw201 by Sodium dodecyl sulfate polyacrylamide gel electrophoresis (SDS-PAGE). Lane M, pre-stained protein ladder; Lane 1, purified Alyw201.

**Figure 6 marinedrugs-18-00189-f006:**
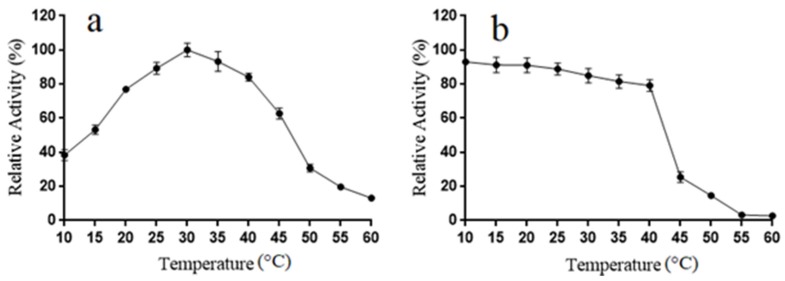
(**a**) Effect of different temperatures on the activity of Alyw201. (**b**) Effect of different temperatures on the stability of Alyw201. Data are given as means ± standard deviation, *n* = 3.

**Figure 7 marinedrugs-18-00189-f007:**
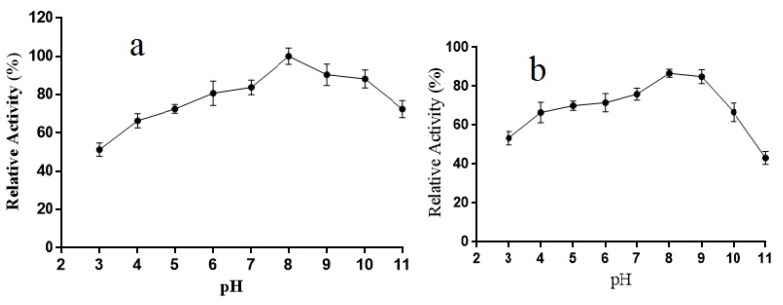
(**a**) Effect of different pH levels on the activity of Alyw201. (**b**) Effect of different pH levels on the stability of Alyw201. Data are given as means ± standard deviation, *n* = 3.

**Figure 8 marinedrugs-18-00189-f008:**
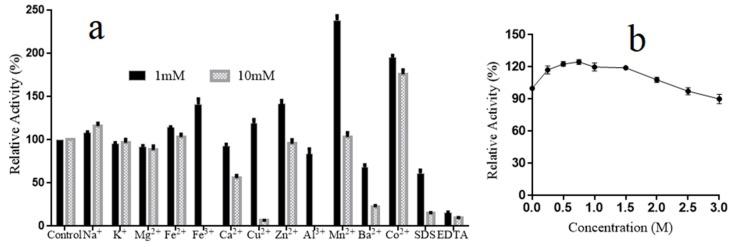
(**a**) Effects of metal ions, EDTA, and SDS on the activity of Alyw201. (**b**) Effects of NaCl on the activity of Alyw201. Data are shown as means ± standard deviation, *n* = 3.

**Figure 9 marinedrugs-18-00189-f009:**
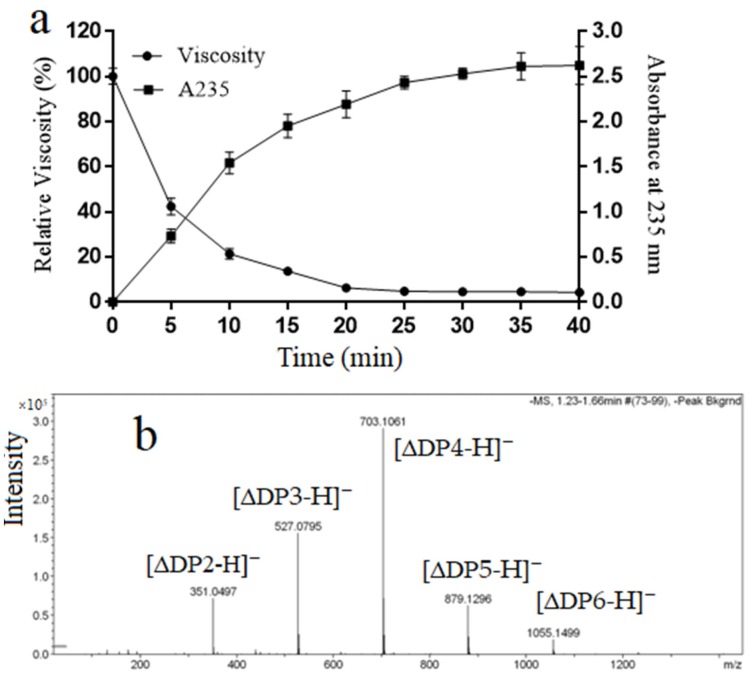
(**a**) Viscosity and absorbance changes during the degradation process. Data are shown as means ± standard deviation, *n* = 3. (**b**) Analysis of the degradation products of Alyw201 by Electrospray ionization mass spectrometry (ESI-MS).

**Table 1 marinedrugs-18-00189-t001:** Comparison of the properties of Aly08 with the reported cold-adapted alginate lyases.

Name	Source	Optimal pH/ Temperature (°C)	Activity at 20 °C	pH-stable Range	Product (DP, Degree of Polymerization)
Alyw201	This study	8.0/35	76.96%	4–10	2–6
TsAly6A	*Thalassomonas* sp. [16]	8.0/35	73.1%	6.6–8.95	2–3
TsAly7B	*Thalassomonas* sp. [15]	8.0/20	100%	7.3–8.6	2–3
AlyPM	*Pseudoalteromonas* sp. [10]	8.0/35	52%	-	1
ZH0-IV	*Sphingomonas* sp. [18]	7.5/35	75%	6.0–9.0	1
AlyGC	*Glaciecola chathamensis* [19]	7.0/30	62.5%	-	1
Algb	*Vibrio* sp. W13 [31]	8.0/30	76%	4–10	2–5
A9m	*Vibrio* sp. A9mT [26]	7.5/30	-	7–10	-
AlgNJU-03	*Vibrio* sp. NJU-03 [24]	7.0/30	62%	6.0–9.0	2–5

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
