# Peer review of "Cloning, Secretory Expression and Characterization of a Unique pH-Stable and Cold-Adapted Alginate Lyase"

_marinedrugs, 2020, doi:10.3390/md18040189_

Round 1
Reviewer 1 Report
The MS described by Wang et al. describes the discovery of a cold-adapted alginate lyase from gut microbiome in abalone. Overall, although the authors did provide sufficient data to convey their findings, the manuscript was written primitively. Materials and Methods did not show enough information for reproducing by the readers of Marine Drugs. The reason for the demand of cold-adapted enzyme is too brief and weak in introduction. At this moment, the MS is not relevant for publication in Marine Drugs. Please reconsider my comments below and rewrite the manuscript.
Comments and questions to the authors:
- Abstract “cold-adapted”
In general, which temperature is defined as “cold” ? The authors used abalones to find the enzyme, and abalone is temperature-sensitive and requires 10-20 ËšC. In this point, optimum temperature of Alyw201 is relatively high.
One more, why is cold-adapted alginate lyase required? The authors used extracted enzyme for total characterizatioin. Maybe, the use in industry would be also enzyme cocktails or something. Basically, industrial enzymes required “thermostable”, therefore, the enzymes are often screened from thermophiles. The authors mentioned the reasons are “low risk of contamination and low energy consumption” (ref #7). However, ref #7 described by Li et al. reported oligosaccharide production by Yarrowia lipolytica strain expressing alginate lyase. I can agree with Li et al. because the culture temperature had better be lower to skip bacterial contamination and for continuous cultivation. But “low risk of contamination and low energy consumption” is not relevant for this study for only 40 min.
- Abstract “food-grade host”
I wondering why the authors stick with the host although they used extracted enzyme. Pharmaceutical production in E. coli has long history and nobody deny it.
- Molecular weight estimated by SDS-PAGE
The authors estimated 38 kDa for recombinant protein harvested from supernatant as a secreted protein.
And they calculated theoretical MW 36.4 kDa plus MW of His-tag (ca. 1 kDa) corresponded to 38 kDa. However, secreted protein means signal peptide was removed. I looked for signal peptide sequence in MS and gene accession number of Alyw201 but they were not described. Who can believe the result unless supported data?
If signal peptide consists of 21 amino acid residues, the authors’ estimation is wrong.
- L148-149. “Compared with the cold-adapted alginate lyases in Table1, Alyw201 showed higher activity at 20 °C, and possesses better thermostability.”
Table 1 did not show the data concerning thermostability. If the authors meant Fig. 6b, it was not mentioned anywhere.
- Fig.8, effect of EDTA
The effect of EDTA in Fig.8 is remarkable. It seems like metalloenzymes. Another reported alginate lyases showed more than 60% of activity remaining. This result should be analyzed further.
- Materials and Methods, enzyme assay
The method of enzyme assay was not described and enzyme unit was not defined here. I looked for the reference # 33 mentioned in L.238, but it was not on the reference list. Assay was performed at 25 ËšC when screened on the plate or at 35 ËšC when assayed with NaCl, chemical compound and metal ions. But optimum temperature is 30 ËšC. What do these mean?
- Materials and Methods
The authors did not describe the Materials and Methods correctly for the readers. This is the critical problems.
“To get coding sequence of alginate lyase, the genomic DNA of strain W2 was sequenced and annotated (Novogene, China).” You may genome sequencing was entrusted with Novogene, however, genome seq accession number or gene accession number must be mentioned in the MS. This is mentioned in Instruction for authors for Marine Drugs. Moreover, annotation method is important for reliability. The authors must correct this point according to the vendors report. How is the homology or e-value to annotate the function?
L.284 “Analysis of Alyw201 reaction products”
When you used ESI-MS, used equipment, column and determination conditions must be described.
Minor comments:
I recommend to rewrite it according to professional English editor who is a senior scientist.
- sequence of tenses is not correct
“it was found that the 16S rDNA sequence is most similar to identified strain Vibrio algivorus (98.05%).”
- 95 “blasting”
BLAST is noun
- never start a sentence with a number. Spell Out Numbers at the Start of a Sentence.
- 233 “106 single colonies”
Author Response
The MS described by Wang et al. describes the discovery of a cold-adapted alginate lyase from gut microbiome in abalone. Overall, although the authors did provide sufficient data to convey their findings, the manuscript was written primitively. Materials and Methods did not show enough information for reproducing by the readers of Marine Drugs. The reason for the demand of cold-adapted enzyme is too brief and weak in introduction. At this moment, the MS is not relevant for publication in Marine Drugs. Please reconsider my comments below and rewrite the manuscript.
Authors’ response:
We would like to thank you for the kindly help to revise this manuscript. We appreciated all the comments and suggestions. The reason for the demand of cold-adapted enzyme has been enriched in introduction.
Comments and questions to the authors:
Abstract “cold-adapted”
In general, which temperature is defined as “cold”? The authors used abalones to find the enzyme, and abalone is temperature-sensitive and requires 10-20 ËšC. In this point, optimum temperature of Alyw201 is relatively high.
Authors’ response:
We appreciate the reviewer’s comments. Cold-adapted enzymes show relatively higher catalytic activities than mesophilic homologs at low temperatures (<25°C), although the optimal temperature maybe higher. This is different from those psychrophilic enzymes.
Generally, most reported alginate lyases showed highest activity at about 40 °C; while the cold-adapted alginate lyases performed highest catalytic activities at less than 35 °C, and commonly performed more than 50% of the highest activity at 20 °C. As described in ref #7, ref #10, and ref #16, the optimal temperatures of the three cold-adapted alginate lyases were 35 °C, 35 °C, and 30 °C, respectively. Actually, the three cold-adapted alginate lyases were also derived from bacteria at low temperatures (<25°C). Thus, a higher optimum temperature of Alyw201 than the environmental temperature was reasonable.
Gao, S.; Zhang, Z.; Li, S.; Su, H.; Tang, L.; Tan, Y.; Yu, W.; Han, F. Characterization of a new endo-type polysaccharide lyase (PL) family 6 alginate lyase with cold-adapted and metal ions-resisted property. Int. J. Biol. Macromol. 2018, 120, 729–735.
Li, S.; Yang, X.; Bao, M.; Wu, Y.; Yu, W.; Han, F. Family 13 carbohydrate-binding module of alginate lyase from Agarivorans sp. L11 enhances its catalytic efficiency and thermostability, and alters its substrate preference and product distribution. FEMS Microbiol Lett. 2015, 362 (10).
Chen, X.; Dong, S.; Xu, F.; Dong, F.; Li, P.; Zhang, X.; Zhou, B.; Zhang, Y.; Xie, B. Characterization of a new cold-adapted and salt-activated polysaccharide lyase family 7 alginate lyase from Pseudoalteromonas sp. SM0524. Front Microbiol. 2016, 7: 1120.
One more, why is cold-adapted alginate lyase required? The authors used extracted enzyme for total characterization. Maybe, the use in industry would be also enzyme cocktails or something. Basically, industrial enzymes required “thermostable”, therefore, the enzymes are often screened from thermophiles. The authors mentioned the reasons are “low risk of contamination and low energy consumption” (ref #7). However, ref #7 described by Li et al. reported oligosaccharide production by Yarrowia lipolytica strain expressing alginate lyase. I can agree with Li et al. because the culture temperature had better be lower to skip bacterial contamination and for continuous cultivation. But “low risk of contamination and low energy consumption” is not relevant for this study for only 40 min.
Authors’ response:
Sorry for the misconception caused by the unclear description. This part was rewritten. In ref #7, the culture temperature was lower to skip bacterial contamination. Importantly, the alginate lyase in Yarrowia lipolytica (ef #7) was also cold-adapted to perform the enzymatic hydrolysis at the culture temperature.
Using alginate as a substrate, thermostable alginate lyases are better candidates. When using seaweed as a substrate with rich fermentable sugars, the reaction time can be more than 10 h. cold-adapted alginate lyases are more preferable, to lower risk of contamination and lower energy consumption.
Sharma S, Horn SJ. Enzymatic Saccharification of Brown Seaweed for Production of Fermentable Sugars. Bioresour Technol, 2016 (213), 155-161.
Florez-Fernandez, N., Torres, M.D., Gonzalez-Munoz, M.J., and Dominguez, H. Recovery of bioactive and gelling extracts from edible brown seaweed Laminaria ochroleuca by non-isothermal autohydrolysis. Food Chem. 2019, 277, 353–361.
Khan, W.; Rayirath, U. P.; Subramanian, S.; Jithesh, M. N.; Rayorath, P.; Hodges, D. M.; Critchley, A. T.; Craigie, J. S.; Norrie, J.; Prithiviraj, B. Seaweed extracts as biostimulants of plant growth and development. J Plant Growth Regul. 2009, 28(4), 386–399.
Abstract “food-grade host”
I wondering why the authors stick with the host although they used extracted enzyme. Pharmaceutical production in E. coli has long history and nobody deny it.
Authors’ response:
Although E. coli is dominant for pharmaceutical production, Y. lipolytica has unique advantages, such as remarkable extracellular secretion, no antibiotic addition, and no endotoxin synthesis. Meaningfully, GRAS (generally recognized as safe) status creates Y. lipolytica as an attractive and environmentally friendly microbial host for the manufacturing of nutraceuticals, fermented food, and dietary supplements.
Molecular weight estimated by SDS-PAGE
The authors estimated 38 kDa for recombinant protein harvested from supernatant as a secreted protein. And they calculated theoretical MW 36.4 kDa plus MW of His-tag (ca. 1 kDa) corresponded to 38 kDa. However, secreted protein means signal peptide was removed. I looked for signal peptide sequence in MS and gene accession number of Alyw201 but they were not described. Who can believe the result unless supported data? If signal peptide consists of 21 amino acid residues, the authors’ estimation is wrong.
Authors’ response:
Sorry for the error. The theoretical MW of Alyw201 without signal peptide is 36.4 kDa; the MW of His-tag was 0.84 kDa. Thus, the theoretical MW of secreted protein should be was 37.2 kDa. Additionally, the discrepancy with the real MW detected by SDS-PAGE maybe attributed to the glycosylation at the site of NRTD.The amino acid sequence of Alyw201 has been submitted with an ID of 2325272. The accession number will be added when received from NCBI.
L148
-149. “Compared with the cold-adapted alginate lyases in Table1, Alyw201 showed higher activity at 20 °C, and possesses better thermostability.”
Table 1 did not show the data concerning thermostability. If the authors meant Fig. 6b, it was not mentioned anywhere.
Authors’ response:
Thanks for your kind recommendation. Fig. 6b has been mentioned.
Fig.8, effect of EDTA
The effect of EDTA in Fig.8 is remarkable. It seems like metalloenzymes. Another reported alginate lyases showed more than 60% of activity remaining. This result should be analyzed further.
Authors’ response:
Thanks for your kind recommendation. The analysis was added.
Materials and Methods, enzyme assay
The method of enzyme assay was not described and enzyme unit was not defined here. I looked for the reference # 33 mentioned in L.238, but it was not on the reference list. Assay was performed at 25 ËšC when screened on the plate or at 35 ËšC when assayed with NaCl, chemical compound and metal ions. But optimum temperature is 30 ËšC. What do these mean?
Authors’ response:
Thanks for your kind recommendation and sorry for the error. The description of enzyme assay and the definition of enzyme unit have been added. Reference # 33 has been inserted. Screening alginate lyase-producing strains was performed at 25 ËšC to balance the strain growth and enzymatic hydrolysis of the plate. By comparing the sizes of the transparent circles around the strains, the relative level of alginate lyase activity at 25 °C could be evaluated. There is no need to proceed the enzyme assay.
Before studying the effects of temperature on Alyw201, the optimum temperature was unknown. The enzyme assay was performed at 35 ËšC with NaCl, according to previous studies.
Materials and Methods
The authors did not describe the Materials and Methods correctly for the readers. This is the critical problems.
“To get coding sequence of alginate lyase, the genomic DNA of strain W2 was sequenced and annotated (Novogene, China).” You may genome sequencing was entrusted with Novogene, however, genome seq accession number or gene accession number must be mentioned in the MS. This is mentioned in Instruction for authors for Marine Drugs. Moreover, annotation method is important for reliability. The authors must correct this point according to the vendors report. How is the homology or e-value to annotate the function?
Authors’ response:
Thanks for your kind recommendation. The amino acid sequence of Alyw201 has been submitted with an ID of 2325272. The accession number will be added when received from NCBI. The annotation method has been added in Materials and Methods.
L.284 “Analysis of Alyw201 reaction products”
When you used ESI-MS, used equipment, column and determination conditions must be described.
Authors’ response:
Thanks for your kind recommendation. The conditions were added.
Minor comments:
I recommend to rewrite it according to professional English editor who is a senior scientist.
Authors’ response:
Thanks for your kind recommendation. The manuscript has been sent for English editing service provide by MDPI.
sequence of tenses is not correct
“it was found that the 16S rDNA sequence is most similar to identified strain Vibrio algivorus (98.05%).”
Authors’ response: Thanks for your kind recommendation. It has been corrected.
95 “blasting”
BLAST is noun
Authors’ response: Thanks for your kind recommendation. It has been corrected.
never start a sentence with a number. Spell Out Numbers at the Start of a Sentence.
233 “106 single colonies”
Authors’ response: Thanks for your kind recommendation. It has been corrected.
Reviewer 2 Report
marine drugs 747559 rev
The manuscript entitled "Cloning, Secretory Expression and Characterization of a Unique pH-Stable and Cold-Adapted Alginate Lyase" addresses a relevant topic regarding the use of active lyases in the degradation of alginate at low temperatures. In general the manuscript is well structured and the experimental part well described. However, I recommend that the authors carry out a thorough review of written English and introduce the corrections that I enumerate below.
Corrections needed
line 20 - "sp." in normal letters, please
line 31 - Brown seaweed (Phaeophyceae)
line 47 - ...different bacteria, ...
line 58 - "sp." in normal letters, please
line 60 - ... host Yarrowia lipolytica (Fungi).
line 68 - ... staining by Lugol's iodine solution...
line 75 - Figure 1. ...staining by Lugol's iodine solution...
Table 1. Thalassomonas sp.
line 201 - ... 35 min.
Author Response
The manuscript entitled "Cloning, Secretory Expression and Characterization of a Unique pH-Stable and Cold-Adapted Alginate Lyase" addresses a relevant topic regarding the use of active lyases in the degradation of alginate at low temperatures. In general the manuscript is well structured and the experimental part well described. However, I recommend that the authors carry out a thorough review of written English and introduce the corrections that I enumerate below.
Authors’ response:
Thanks for your kind recognition and recommendation. The manuscript has been sent for English editing service provide by MDPI.
Corrections needed
line 20 - "sp." in normal letters, please
Authors’ response: Thanks for your kind recommendation. It has been corrected.
line 31 - Brown seaweed (Phaeophyceae)
Authors’ response: Thanks for your kind recommendation. It has been corrected.
line 47 - ...different bacteria, ...
Authors’ response: Thanks for your kind recommendation. It has been corrected.
line 58 - "sp." in normal letters, please
Authors’ response: Thanks for your kind recommendation. It has been corrected.
line 60 - ... host Yarrowia lipolytica (Fungi).
Authors’ response: Thanks for your kind recommendation. It has been corrected.
line 68 - ... staining by Lugol's iodine solution...
Authors’ response: Thanks for your kind recommendation. It has been corrected.
line 75 - Figure 1. ...staining by Lugol's iodine solution...
Authors’ response: Thanks for your kind recommendation. It has been corrected.
Table 1. Thalassomonas sp.
Authors’ response: Thanks for your kind recommendation. It has been corrected.
line 201 - ... 35 min.
Authors’ response: Thanks for your kind recommendation. It has been corrected.
Reviewer 3 Report
The manuscript by Wang et al. present the identification of an Alginate Lyase that has improved pH stability and importantly a high activity at ambient temperature. The work undertaken follow a very logical structure to characterize the Alginate lyase. Nevertheless, the results lack any statistic analysis to see whether the differences observed in activity under different pH conditions (Fig. 7), or in the presence of metal ions (Fig. 8) are significant or not.
Minors comments:
Lane 11: remove b from "bLaboratory"
Lane 67: Define ASC
Lane 73: Give SD for the value or do not show it at all
Figure 2: Explain what the number on the branches indicate or remove them
Lane 97: Change to "a phylogenetic tree was constructed"
Lane 108: How have they been verified?
Lane 112: Correct "maybe"
Lane 119: "were marked by black boxes"
Lane 125: Define GPPB
Lane 132: Is this significant?
Lane 144-145: Change to: However, most of the activity was lost at temperatures above 45C.
Lane 149: Explain better because based on the table AlyPM is better
Figure 6 legend: More details about the experiment should be given here.
Lane 192: Needs english correction
Figure 9a: Label X-axis
Figure 9b: Intensity (Y-axis)
Lane 216: Change to: Alyw201 activity was also independent of NaCl concentration. The degraded products were...
Lane 217: Remove: was
Lane 247: Provide accession number if it has been annotated
Lane 248: English
Author Response
The manuscript by Wang et al. present the identification of an Alginate Lyase that has improved pH stability and importantly a high activity at ambient temperature. The work undertaken follow a very logical structure to characterize the Alginate lyase. Nevertheless, the results lack any statistic analysis to see whether the differences observed in activity under different pH conditions (Fig. 7), or in the presence of metal ions (Fig. 8) are significant or not.
Authors’ response:
Thanks for your kind recognition and recommendation. The data in Fig. 7 or Fig. 8 was not comparison between groups, so statistical analysis about the significance cannot be necessary. Moreover, in previous reported studies, no statistical analysis about the significance was performed.
Minors comments:
Lane 11: remove b from "bLaboratory"
Authors’ response: Thanks for your kind recommendation. It has been removed.
Lane 67: Define ASC
Authors’ response: Thanks for your kind recommendation. It has been defined
Lane 73: Give SD for the value or do not show it at all
Authors’ response: Thanks for your kind recommendation. It has been removed.
Figure 2: Explain what the number on the branches indicate or remove them
Authors’ response: Thanks for your kind recommendation. It has explained.
Lane 97: Change to "a phylogenetic tree was constructed"
Authors’ response: Thanks for your kind recommendation. It has been corrected.
Lane 108: How have they been verified?
Authors’ response: Thanks for your kind recommendation. It has been corrected.
Lane 112: Correct "maybe"
Authors’ response: Thanks for your kind recommendation. It has been corrected.
Lane 119: "were marked by black boxes"
Authors’ response: Thanks for your kind recommendation. It has been corrected.
Lane 125: Define GPPB
Authors’ response: Thanks for your kind recommendation. It has been defined.
Lane 132: Is this significant?
Authors’ response: Thanks for your kind recommendation. It was significant, but we think there is no need to show it.
Lane 144-145: Change to: However, most of the activity was lost at temperatures above 45C.
Authors’ response: Thanks for your kind recommendation. It has been corrected.
Lane 149: Explain better because based on the table AlyPM is better
Authors’ response: No, Alyw201 in this study was much better than AlyPM obviously.
Figure 6 legend: More details about the experiment should be given here.
Authors’ response: Thanks for your kind recommendation. More details are present in Materials and Methods.
Lane 192: Needs english correction
Authors’ response: Thanks for your kind recommendation. The manuscript has been sent for English editing service provide by MDPI.
Figure 9a: Label X-axis
Authors’ response: Thanks for your kind recommendation. It has been corrected.
Figure 9b: Intensity (Y-axis)
Authors’ response: Thanks for your kind recommendation. It has been corrected.
Lane 216: Change to: Alyw201 activity was also independent of NaCl concentration. The degraded products were...
Authors’ response: Thanks for your kind recommendation. It has been corrected.
Lane 217: Remove: was
Authors’ response: Thanks for your kind recommendation. The word should be remained according to the English editing service.
Lane 247: Provide accession number if it has been annotated
Authors’ response: Thanks for your kind recommendation. It has been corrected.
Lane 248: English
Authors’ response: Thanks for your kind recommendation. It has been corrected.
Round 2
Reviewer 1 Report
Some of the questions I arose meet the answers. But remaining answers did not meet my points.
1. Once again, the case likewise Li et al. oligosaccharide production during cultivation is reasonable. Newly added reference #4 employed commercial enzyme cocktails to produce sugars at 50 ËšC. Please realize the optimum temperature is up to applications.
2. As the authors answered, yeasts modify the sugars and the glycosylation sometime affect the structure and inhibit docking with substrate. Therefore, this bacterial enzyme had better be expressed in E.coli. This is my point.
3. Effect of EDTA,
General alginate lyases do not have metal binding site. Therefore, the avtivity was remained after EDTA treatment. However, the enzyme in this MS got critical inhibition by EDTA. This is very unique if the assay was correctly done. I did not require such a discussion based on the sequence but biochemistry.
4. Sequence chemistry is not mentioned.
Author Response
- Once again, the case likewise Li et al. oligosaccharide production during cultivation is reasonable. Newly added reference #4 employed commercial enzyme cocktails to produce sugars at 50 ËšC. Please realize the optimum temperature is up to applications.
Authors’ response:
Thanks for your kind recommendation. Lower risk of contamination and lower energy consumption can be achieved during cultivation of recombinant strain with cold-adapted alginate lyases. The improper description has been corrected.
- As the authors answered, yeasts modify the sugars and the glycosylation sometime affect the structure and inhibit docking with substrate. Therefore, this bacterial enzyme had better be expressed in E.coli. This is my point.
Authors’ response:
Thanks for your kind recommendation. In this study, it was indicated that this bacterial enzyme had better be expressed in E. coli, although Y. lipolytica possesses GRAS status. The comments were added in Section 2.3.
- Effect of EDTA,
General alginate lyases do not have metal binding site. Therefore, the avtivity was remained after EDTA treatment. However, the enzyme in this MS got critical inhibition by EDTA. This is very unique if the assay was correctly done. I did not require such a discussion based on the sequence but biochemistry.
Authors’ response:
Thanks for your kind recommendation. The revised discussion has been supplemented. We confirmed the results are right. Critical inhibition of by EDTA has been also found in several reported alginate lyases. In ref #19, when metal chelator EDTA was added to the reaction mixture, the activity of alginate lyase AlyGC decreased with the increase of EDTA concentration, and 0.2 mM EDTA completely abolished the enzyme activity. A combined structural and mutational analysis indicates that Ca2+, which is far away from the active center, is involved in catalysis and plays an important role in stabilizing the structure. In this study, it was also suspected that EDTA chelates the metal ions crucial for the function and structure of Alyw201.
- Sequence chemistry is not mentioned.
Authors’ response: The chemical characteristic of the sequence was added in Section 4.5.

Reviewer 2 Report
The authors made the corrections indicated by the reviewers, so I believe that the manuscript can be accepted for publication
Author Response
The authors made the corrections indicated by the reviewers, so I believe that the manuscript can be accepted for publication
Authors’ response: Thanks for your kind recognition.
